# Oxy210 Inhibits Hepatic Expression of Senescence-Associated, Pro-Fibrotic, and Pro-Inflammatory Genes in Mice During Development of MASH and in Hepatocytes In Vitro

**DOI:** 10.3390/cells14151191

**Published:** 2025-08-02

**Authors:** Feng Wang, Simon T. Hui, Frank Stappenbeck, Dorota Kaminska, Aldons J. Lusis, Farhad Parhami

**Affiliations:** 1MAX BioPharma Inc., Santa Monica, CA 90404, USA; fwang@maxbiopharma.com (F.W.); fstappenbeck@maxbiopharma.com (F.S.); 2Department of Medicine, Division of Cardiology, David Geffen School of Medicine, University of California, Los Angeles, CA 90095, USA; sthui@mednet.ucla.edu (S.T.H.); dkaminska@mednet.ucla.edu (D.K.); jlusis@mednet.ucla.edu (A.J.L.)

**Keywords:** MASH, oxysterols, Oxy210, senescence, fibrosis

## Abstract

Background: Senescence, a state of permanent cell cycle arrest, is a complex cellular phenomenon closely affiliated with age-related diseases and pathological fibrosis. Cellular senescence is now recognized as a significant contributor to organ fibrosis, largely driven by transforming growth factor beta (TGF-β) signaling, such as in metabolic dysfunction-associated steatohepatitis (MASH), idiopathic pulmonary fibrosis (IPF), chronic kidney disease (CKD), and myocardial fibrosis, which can lead to heart failure, cystic fibrosis, and fibrosis in pancreatic tumors, to name a few. MASH is a progressive inflammatory and fibrotic liver condition that has reached pandemic proportions, now considered the largest non-viral contributor to the need for liver transplantation. Methods: We previously studied Oxy210, an anti-fibrotic and anti-inflammatory, orally bioavailable, oxysterol-based drug candidate for MASH, using APOE*3-Leiden.CETP mice, a humanized hyperlipidemic mouse model that closely recapitulates the hallmarks of human MASH. In this model, treatment of mice with Oxy210 for 16 weeks caused significant amelioration of the disease, evidenced by reduced hepatic inflammation, lipid deposition, and fibrosis, atherosclerosis and adipose tissue inflammation. Results: Here we demonstrate increased hepatic expression of senescence-associated genes and senescence-associated secretory phenotype (SASP), correlated with the expression of pro-fibrotic and pro-inflammatorygenes in these mice during the development of MASH that are significantly inhibited by Oxy210. Using the HepG2 human hepatocyte cell line, we demonstrate the induced expression of senescent-associated genes and SASP by TGF-β and inhibition by Oxy210. Conclusions: These findings further support the potential therapeutic effects of Oxy210 mediated in part through inhibition of senescence-driven hepatic fibrosis and inflammation in MASH and perhaps in other senescence-associated fibrotic diseases.

## 1. Introduction

Metabolic dysfunction-associated steatohepatitis (MASH) is a progressive inflammatory and fibrotic liver condition, initiated by excess deposition of lipids in liver tissue, such as triglycerides and cholesterol. The initial steatotic liver disease, also known as metabolic dysfunction-associated steatotic liver disease (MASLD), can advance toward MASH driven by chronic unresolving inflammation, resulting in progressive and increasingly irreversible liver damage, i.e., fibrosis, cirrhosis, and eventually liver failure. Fibrosis itself, which could be viewed broadly as a marker of tissue aging, is characterized by aberrant inflammatory and wound healing responses that can affect major visceral organs, such as liver, lung, and kidney, or promote other conditions, such as atherosclerosis and type 2 diabetes [1]. In recent decades, the incidence of MASLD/MASH has been steadily increasing in large part due to the obesity epidemic [2].

Cellular senescence, a state of permanent cell cycle arrest, has emerged as a conceptual keystone in our modern understanding of aging and disease, transversing various fields, such as cancer, inflammation, fibrosis, metabolism, and wound healing [3]. Cellular senescence is now recognized as a significant contributor to the development of MASLD/MASH and other fibrotic diseases that are often age-related, as well as age-related diseases more generally [4]. Examples include neurodegenerative diseases, such as Alzheimer’s and Parkinson’s disease; musculoskeletal disorders, such as osteoarthritis and osteoporosis, and diverse fibroproliferative disorders, including atherosclerosis, MASH, and idiopathic pulmonary fibrosis (IPF). Atherosclerosis and MASH partially overlap with other cardiometabolic disorders, such as MASLD, metabolic syndrome, obesity, and type 2 diabetes that are also associated with senescence [5].

Senescence is linked to a distinct set of changes in cellular behavior, known as the senescence-associated secretory phenotype (SASP). This complex phenotype is characterized by morphological changes, altered gene expression and cellular metabolism, production and secretion of pro-inflammatory and pro-fibrotic molecules that activate neighboring cell populations, such as myofibroblasts, hepatic stellate cells (HSCs), and macrophages, and drive the progression of fibrotic disease. In the liver, senescence occurs in subsets of hepatocytes [6], the primary and most abundant functional cells of the liver, although other less abundant liver cell types, such as Kupffer cells [7] and HSCs may also be affected [8].

The buildup of senescent hepatocytes is likely to promote hepatic fat accumulation and steatosis during the onset of MASH, as suggested by the close correlation between hepatic fat accumulation and markers of hepatocyte senescence observed in vitro and vivo [9]. Conversely, the elimination of senescent liver cells, by genetic or pharmacological means, can reduce overall hepatic steatosis, whereas experimentally induced hepatocyte senescence promotes fat accumulation in animal models of the disease [9]. A likely mechanistic scenario may be that senescent hepatocytes activate nearby (non-senescent) HSCs, Kupffer cells, and macrophages via paracrine signaling from SASP-based secretions. When activated, HSCs are the primary source of myofibroblasts and TGF-β in the liver and drive the progression of liver fibrosis in MASH [10], while activated Kupffer cells and macrophages drive inflammatory responses and oxidative stress during MASH development [11]. Kupffer cells and macrophages are also a significant source of TGF-β in the (pre-fibrotic) liver and promote HSC activation and fibrogenesis during the development of MASH [12]. Senescent HSCs are often observed in cirrhotic livers during late-stage MASH, when the proliferative and collagen-producing capacity of HSCs gives way to increased inflammatory properties [13]. TGF-β is considered a leading inducer of not only fibrosis but also senescence in fibroblasts, hepatocytes, and endothelial cells [1].

We previously studied Oxy210, an orally bioavailable, oxysterol-based, anti-fibrotic and anti-inflammatory drug candidate for MASH, using humanized hyperlipidemic APOE*3-Leiden.CETP mice that develop MASH symptoms when fed a high fat, high cholesterol “Western” diet (WD) over 16 weeks [14]. Mice treated with Oxy210 showed ameliorated hepatic hallmarks of MASH, including inflammation and fibrosis [15]. Outside of the liver, Oxy210 reduced atherosclerosis [16] and adipose tissue inflammation [17] in these mice while reducing total and unesterified cholesterol and inflammatory cytokine levels in circulation. In the present report, we investigate the effects of Oxy210 on markers of cellular senescence and fibrosis in liver tissue harvested from APOE*3-Leiden.CETP mice during progression of the disease from steatosis to MASH in 16 weeks. In addition, we examine the effects of Oxy210 in vitro using HepG2 human hepatocytes treated with TGF-β to induce the expression of senescence genes. We demonstrate that the expression of senescence-related factors significantly increases during the development of MASH and correlates with the increased expression of pro-fibrotic factors. Furthermore, administration of Oxy210 inhibits the increased expression of these factors as early as 4 weeks through the 16-week period. In HepG2 cells, Oxy210 treatment inhibits TGF-β induced expression of senescence-associated genes.

## 2. Materials and Methods

### 2.1. Cell Culture and Reagents

HepG2 cells were obtained from ATCC and cultured in EMEM containing 10% FBS and antibiotics penicillin and streptomycin. rhTGF-β1 was obtained from R&D Systems. Synthesis and characterization of Oxy210 (3*S*,8*S*,9*S*,10*R*,13*S*,14*S*,17*S*)-17-((*R*)-2-hydroxy-4-(pyridin-3-yl)butan-2-yl)-10,13-dimethyl-2,3,4,7,8,9,10,11,12,13,14,15,16,17-tetradecahydro-1H-cyclopenta[a]phenanthren-3-ol) was performed at MAX BioPharma in three synthetic steps as previously described [15]. Briefly, pregnenolone was condensed with nicotinaldehyde to an enone which was reduced to a ketone via hydrogenation using Lindlar’s catalyst. The ketone was reacted with methyl lithium to afford the 20(*R*)-tertiary alcohol, Oxy210. The crude product was purified by chromatography on silica.



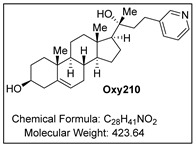



### 2.2. RNA Isolation and Quantitative RT-PCR

Total RNA was extracted from the cells with the RNeasy Plus Mini Kit from Qiagen (Hilden, Germany) according to the manufacturer’s instructions. The livers of the mice were homogenized in Qiazol (Qiagen) and were separated into aqueous and organic phases by centrifugation. Total RNA in aqueous phase was extracted with the RNeasy Plus Universal Mini Kit from Qiagen. One microgram of RNA was reverse-transcribed using an iScript Reverse Transcription Supermix from Bio-Rad (Hercules, CA, USA), to make single-stranded cDNA. Quantitative PCR was performed with SYBR Select Master Mix for CFX (Applied Biosystems, Waltham, MA, USA) using a CFX Connect Real-Time PCR System (Bio-Rad). All PCR samples were prepared in triplicate wells in a 96-well plate. Fold changes in gene expression were calculated using the ΔΔCt method. Sequences of primers used are listed in Appendix A.

### 2.3. Animal Studies

The breeding and characterization of transgenic mice expressing human CETP and the human APOE*3-Leiden (E3L) were described previously [14]. To generate mice for the fibrosis studies, male C57BL/6J mice carrying both transgenes were bred to BXD19/TyJ females. F1 progeny carrying both transgenes were used for the studies. Animals were maintained on a 12 h light–dark cycle with ad libitum access to food and water. All diets were formulated by Research Diets, Inc. (New Brunswick, NJ, USA). Control mice were fed a WD (33 kcal% fat from cocoa butter and 1% cholesterol, Research Diets, cat# D10042101) for 16 weeks, whereas Oxy210-treated mice were fed a WD supplemented with Oxy210 at 4 mg/g of food. Food consumption was monitored, and there were no significant differences in body weights at the end of the study, except for a small but significant reduction in body weights in the 4 mg/g cohort [15]. The underlying driver of the reduction in weight gain is currently unknown; however, it is not likely to be caused by toxicity. No evidence of toxicity was noted during the 16-week treatment with Oxy210 and mice exhibited normal behavior, grooming, eating, and physical activity. Necropsies showed no evidence of Oxy210 toxicity to any organs. All animal work was approved by the UCLA Animal Research Committee, the IACUC.

### 2.4. Statistical Analysis

Statistical analyses were performed using the StatView 5 program (SAS Institute, Cary, NC, USA). All *p*-values were calculated using ANOVA and Fisher’s projected least significant difference (PLSD) significance test. A value of *p* < 0.05 was considered significant.

## 3. Results

Numerous studies have connected aberrant TGF-β and Hh signaling to liver fibrosis and MASH [18,19], as well as fibrotic disorders in other organs, such as lung and kidney [20]. We have previously characterized Oxy210 as an inhibitor of TGF-β and Hh signaling, in vitro, in human primary HSCs, and in vivo, in mouse liver tissue. We have also reported on anti-inflammatory properties of Oxy210, conveyed via inhibition of Toll-Like Receptor (TLR)2 (TLR2), TLR4, and AP-1 signaling, observed in vitro in human and murine macrophages and aortic endothelial cells, and in vivo, in mouse plasma, liver and adipose tissue, and the arterial wall. In addition, Oxy210 displays significant cholesterol-lowering effects in APOE*3-Leiden.CETP mice on WD [15,16]. Given the reported role of TGF-β signaling as a major driver of senescence and fibrosis in disease and aging [21], and the role of TLR signaling in mediating the pro-inflammatory properties of senescent cells [22], we decided to investigate the effect of Oxy210 on the expression of pro-fibrotic, pro-inflammatory, senescence-associated and SASP genes in liver tissue of the APOE*3-Leiden.CETP mice during MASH development. WD was administered to APOE*3-Leiden.CETP mice over a time course of 16 weeks during which hallmarks of MASH develop [14]. The expression levels of pro-fibrotic genes *Tgf-b1*, *Col1a1*, *Ctgf*, *Acta2*, *Ptch1*, *Gli1*, and *Spp1*, pro-inflammatory genes, *Ccl2*, *Il-1b*, *Il-6*, and *Tnf-a*, and senescence-associated and SASP genes, NADPH oxidase 2 and 4, *P21*, *P16*, *P53*, *Pai-1*, and *Chop*, were measured at 0, 4, 8, 12 and 16 weeks after placing the mice on WD. In WD control mice, significant increases in gene expression were recorded over 16 weeks and these increases were significantly reduced in animals treated with Oxy210, often to levels seen at the beginning of the WD intervention period.

### 3.1. Gene Expression in Liver Tissue of the APOE*3-Leiden.CETP Mice During MASH Development

#### 3.1.1. Senescence-Associated, SASP, and Pro-Inflammatory Genes (Figure 1, Figure 2 and Figure 3)

P21, also known as CDKN1A, and P15, or P15INK4B, are cyclin-dependent kinase inhibitors that play a role in senescence by inhibiting the activity of cyclin-dependent kinase (CDK) complexes, such as CDK2, CDK1, and CDK4/6 complexes [23,24]. P16, or p16INK4a, also a cell cycle inhibitor, acts primarily through the P16-pRB pathway [25]. P53, also known as tumor protein P53, is a tumor suppressor protein that plays a crucial role in regulating cell cycle progression and senescence [26]. In response to various stressors, P53 and the cell cycle inhibitors, P21, P15 and P16, can work cooperatively or independently to affect senescence [26,27,28]. Activation, including upregulated expression at the RNA level and post-translational modification of P53 [29], and increased gene expression of *P21*, *P15* and *P16* have been considered as markers of cellular senescence in MASH and other fibrotic and chronic inflammatory diseases [30,31]. As shown in Figure 1, our findings show such increases in the hepatic gene expression for *p21*, *p15*, *p16* and *p53*, over the 16-week time course. Oxy210 significantly reduced the increased hepatic gene expression for *p21*, *p15*, and *p53*, suggesting that Oxy210 treatment may reduce a tendency toward cell cycle arrest and senescence in mouse liver tissue associated with the development of MASH. The expression of *p16* was significantly reduced by Oxy210 treatment in weeks 12 and 16 although an inhibitory trend was also seen at other time points.

**Figure 1 cells-14-01191-f001:**
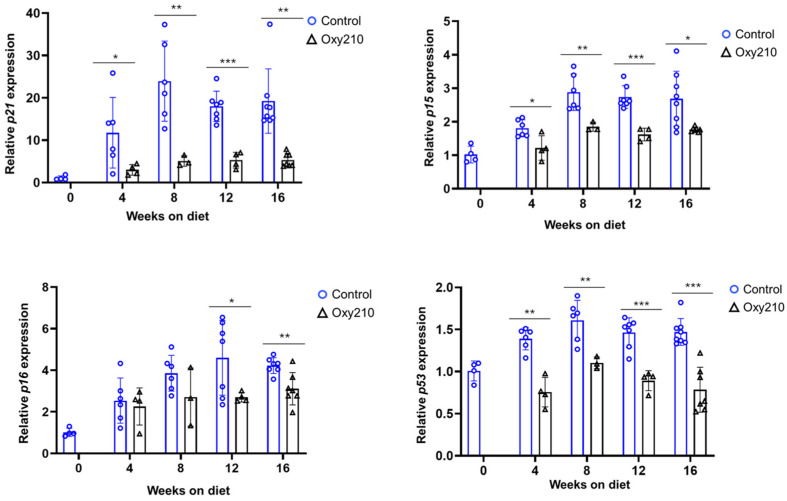
Sixteen-week time course of hepatic senescence-associated gene expression and inhibition by Oxy210 in APOE*3-Leiden.CETP mice. Expression of *p21*, *p15*, *p16* and *p53* in the livers from control and Oxy210-treated mice was measured by qPCR and normalized to the level of the housekeeping gene *Rpl4*. Relative gene expression levels are presented as mean ± SD (* *p* < 0.05, ** *p* < 0.01 and *** *p* < 0.001 vs. Control).

Plasminogen Activator Inhibitor-1 (PAI-1) is a protein that is both a marker and a mediator of cellular senescence [32], partially by inhibiting P53 degradation via proteasomes. C/EBP homologous protein (CHOP) is a transcription factor that is considered a marker of cellular senescence, particularly in the context of fibrosis [33]. As shown in Figure 2, we found increases in the hepatic gene expression for *Pai-1* and *Chop* that were significantly inhibited by Oxy210 during disease progression, consistent with a reduced tendency toward senescence in mouse liver tissue produced by Oxy210 treatment. Oxidative stress, caused by an imbalance between reactive oxygen species (ROS) production and antioxidant defenses, is a known driver of vascular senescence and MASH [34]. As part of the cellular machinery devoted to redox homeostasis, NADPH oxidases (Nox) 2 and 4, members of the Nox family, can generate ROS and trigger various cellular stress responses that ultimately entail cell cycle arrest, inflammation, and fibrosis [35]. As shown in Figure 2, hepatic expression of *Nox2*, but not *Nox4* (Appendix A), was increased over the 16-week time course. Oxy210 significantly reduced the hepatic Nox2 gene expression, suggesting that Oxy210 treatment may result in antioxidant effects.

**Figure 2 cells-14-01191-f002:**
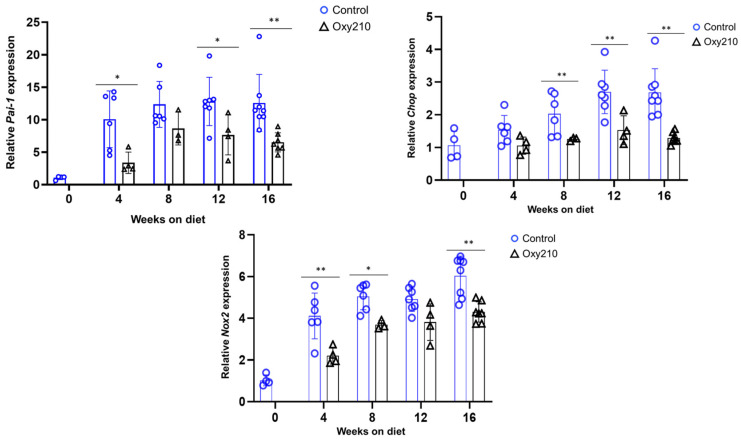
Sixteen-week time course of hepatic SASP gene expression and inhibition by Oxy210 in APOE*3-Leiden.CETP mice. Expression of *Pai-1*, *Chop*, and *Nox2* in the livers from control and Oxy210-treated mice was measured by qPCR and normalized to the level of the housekeeping gene *Rpl4*. Relative gene expression levels are presented as mean ± SD (* *p* < 0.05 and ** *p* < 0.01 vs. Control).

Chronic inflammation causes progressive liver damage, and the progression of MASH and other fibrotic conditions is characterized by cycles of recurring injury, non-resolving inflammation and aberrant wound healing responses. As shown in Figure 3, an increase in the hepatic expression of inflammatory genes *Ccl2, Il-1b*, *Il-6*, and *Tnf-a* between weeks 0 and 8 was found, leveling off after week 12. Oxy210 substantially dampened these increases, particularly between weeks 0 and 8, consistent with its anti-inflammatory properties. It appears that in this model the sequential occurrence of inflammation followed by fibrosis observed in humans is also represented [36,37].

**Figure 3 cells-14-01191-f003:**
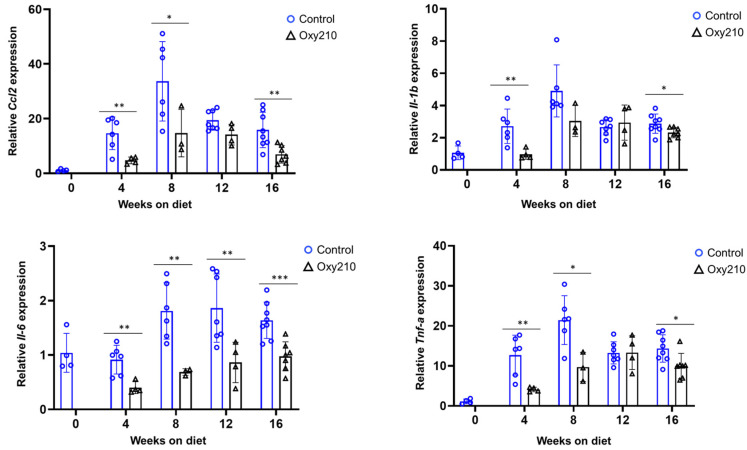
Sixteen-week time course of hepatic inflammatory gene expression and inhibition by Oxy210 in APOE*3-Leiden.CETP mice. Expression of *Ccl2*, *Il-1b*, *Il-6*, and *Tnf-a* in the livers from control and Oxy210-treated mice was measured by qPCR and normalized to the level of the housekeeping gene *Rpl4*. Relative gene expression levels are presented as mean ± SD (* *p* < 0.05, ** *p* < 0.01 and *** *p* < 0.001 vs. Control).

#### 3.1.2. Pro-Fibrotic TGF-β and Hh Target Genes (Figure 4 and Figure 5)

TGF-β is a central mediator of fibrosis in almost all fibrotic diseases, driving extracellular matrix (ECM) deposition and myofibroblast activation [38]. Accordingly, increased expression of TGF-β target genes is closely linked to the progression of fibrosis in many different tissues, including liver tissues [39]. TGF-β signaling is also linked to increased occurrence of cellular senescence in hepatocytes and other liver cells during MASH [21,40]. As shown in Figure 4, our findings recapitulated increased hepatic expression of TGF-β target genes, *Tgf-b1*, *Ctgf*, and *Acta2*, between weeks 0 and 8, leveling off for weeks 12–16. Expression of *Col1a1* appears to be increasing throughout the study. Oxy210 treatment dampened these increases significantly, consistent with its anti-fibrotic properties. Hedgehog (Hh) signaling is crucially involved in cell differentiation and embryonic development but also plays a role in adult tissue homeostasis and repair. Dysregulation of Hh signaling can occur in various disease states, including fibrosis, cancer, and age-related diseases. For many fibrotic conditions, Hh and TGF-β signals drive the activation of myofibroblast cells, specifically HSCs in the case of MASH. Increased amounts of Hh and TGF-β ligands have been detected in human MASH livers, compared to healthy controls, consistent with increased signaling during the development and progression of MASH [41,42]. The present study recapitulates this outcome, as shown in Figure 5: APOE*3-Leiden.CETP mice on WD show increased hepatic expression of Hh target genes, *Ptch1*, *Gli1*, *and Spp1*, particularly for weeks 8–12. Oxy210 dampened these increases significantly, lowering their expression to near baseline levels (week 0).

**Figure 4 cells-14-01191-f004:**
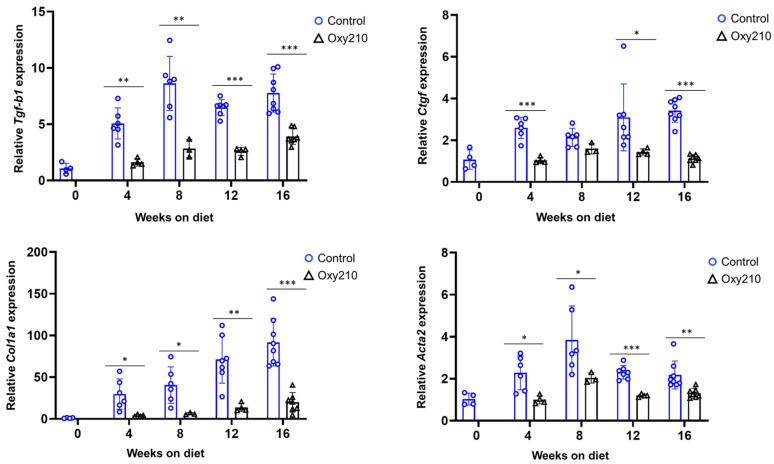
Sixteen-week time course of hepatic TGF-β target gene expression and inhibition by Oxy210 in APOE*3-Leiden.CETP mice. Expression of *Tgf-b1*, *Col1a1*, *Ctgf*, and *Acta2* in the livers from control and Oxy210-treated mice was measured by qPCR and normalized to the level of the housekeeping gene *Rpl4*. Relative gene expression levels are presented as mean ± SD (* *p* < 0.05, ** *p* < 0.01 and *** *p* < 0.001 vs. Control).

**Figure 5 cells-14-01191-f005:**
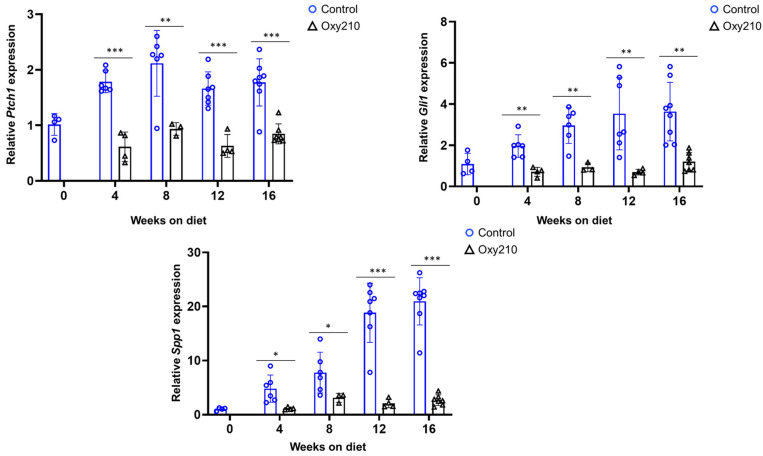
Sixteen-week time course of hepatic Hedgehog target gene expression and inhibition by Oxy210 in APOE*3-Leiden.CETP mice. Expression of *Gli1*, *Ptch1*, and *Spp1* in the livers from control and Oxy210-treated mice was measured by qPCR and normalized to the level of the housekeeping gene *Rpl4*. Relative gene expression levels are presented as mean ± SD (* *p* < 0.05, ** *p* < 0.01 and *** *p* < 0.001 vs. Control).

### 3.2. Effect of Oxy210 on the Expression of Senescence-Associated and Secretory Phenotype (SASP) Genes in Human HepG2 Cells (Figure 6)

We next sought to replicate the effects observed in mouse liver tissue in human HepG2 hepatocyte cells in vitro. In the context of MASH, senescence can be induced by various stress factors such as inflammatory insults, oxidative stress, mitochondrial dysfunction, and dysregulated signaling pathways. Among the latter, aberrant TGF-β signaling stands out as a driver of senescence and fibrosis [1,21]. As shown in Figure 6, HepG2 cells treated with TGF-β ligand showed increased expression of some, but not all, of the senescence markers found to be elevated in liver tissue during the development of MASH. Expression levels of *P21*, *P15*, *TNFA*, and *TGFB1* genes were significantly enhanced by TGF-β and inhibited by Oxy210 treatment to or below baseline levels.

**Figure 6 cells-14-01191-f006:**
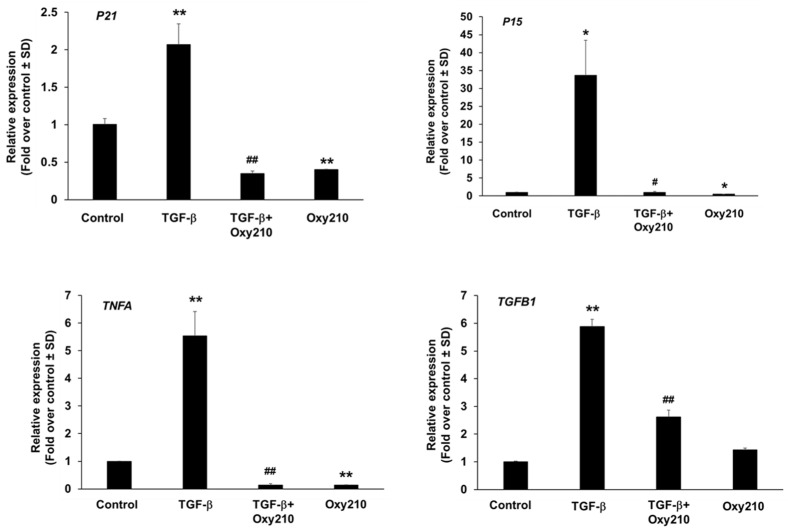
Inhibition of TGF-β-induced expression of senescence-associated genes in HepG2 cells by Oxy210. HepG2 cells were treated overnight in EMEM without FBS followed by treatment with Oxy210 at 10μM in EMEM containing 1% FBS for 24 h and then treated with 50 ng/mL of TGF-β in the absence or presence of Oxy210 for 48 h. RNA was extracted and analyzed by Q-RT-PCR for the expression of the genes as indicated and normalized to *PMSB4* expression. Data from a representative experiment are reported as the mean of triplicate determinations ± SD (## *p* < 0.01 vs. TGF-β; # *p* < 0.05 vs. TGF-b; ** *p* < 0.01 vs. Control; * *p* < 0.05 vs. Control).

## 4. Discussion

In this report, we explored the effects of WD and Oxy210 on the hepatic expression of senescence markers that may originate, at least in significant part, from a subset of hepatocytes that have entered senescence during the development of MASH. We found significant inhibition of senescence markers in mice treated with Oxy210 in parallel with inhibition of pro-fibrotic and pro-inflammatory factors. Our in vitro model for hepatocyte senescence, human HepG2 cells treated with TGF-β, partially confirms these findings, with at least four markers that were induced by TGF-β and significantly inhibited by Oxy210 treatment. While the in vitro experiments in HepG2 cells employed TGF-β to induce senescence, our in vivo model is WD-induced, i.e., stressor induced, suggesting that other factors, for example, oxidative stress and lipotoxicity, in addition to TGF-β signaling could be responsible for producing senescence in the mouse liver tissue. Although the in vitro inhibitory effects of Oxy210 on senescence in HepG2 cells were examined in the context of TGF-β, at this time it is not known whether Oxy210 can inhibit senescence induced by other stressors independent of TGF-β signaling. This is, however, a likely scenario given the rather robust in vivo inhibitory effects of Oxy210 on hepatic senescence markers, and the presence of other stressors that can drive senescence in our mouse model of MASH.

As previously reported, cellular responses to Oxy210 in the presence of TGF-β were found to be cell-type-specific, depending on whether epithelial, endothelial, macrophage, or fibroblasts were studied. Oxy210 inhibits pro-fibrotic and pro-inflammatory responses to TGF-β in epithelial, endothelial and fibroblasts, but it does not inhibit anti-inflammatory responses to TGF-β in macrophages [17,20]. Current results in HepG2 cells, an epithelial cell line, confirm this trend.

Although senescence serves important physiological functions, the accumulation of senescent cells beyond a certain threshold can activate age-related pathological changes. In healthy adults, around 3–7% of hepatocytes are estimated to be senescent, suggesting non-pathological roles of senescent liver cells, but the percentage of senescent hepatocytes can increase significantly during liver disease, such as MASH or cirrhosis [43]. Even a small excess of senescent cells can significantly contribute to MASH or other pathologies due to their secretion of pro-inflammatory and pro-fibrotic factors [44].

Cellular senescence can be divided into different subtypes, such as replicative senescence and stress-induced premature senescence, among others. While the former is age-related and caused by telomere shortening, the latter can be induced by various stress factors, including drugs, such as during cancer treatment, or, in the context of MASH, by inflammatory and lipotoxic insults, oxidative stress, mitochondrial dysfunction, and dysregulated signaling pathways. It is noteworthy that aberrant TGF-β signaling is a major driver of senescence and fibrosis [1,21]. In addition to irreversible cell cycle withdrawal, senescent cells typically exhibit common characteristics in morphology and SASP [45]. SASP characteristics may change over time, from immunosuppressive and pro-fibrotic properties (dominated by TGF-β signaling) to pro-inflammatory properties (dominated by IL-1β, IL-6, and IL-8 signaling). As senescent cells persist and accumulate in damaged tissues, their SASP can induce significant pro-inflammatory and pro-fibrotic effects both on neighboring cells in the affected tissue (paracrine signaling), and even systemically, at distant sites [46]. In the liver, senescence in hepatocytes entails the secretion of SASP factors that activate HSCs and trigger inflammation in a paracrine manner. While senescence can also occur in HSCs; paradoxically, only a subset of senescent HSCs appears to contribute to MASH progression [8], likely by worsening pro-inflammatory liver injury, while other subsets of senescent HSCs may ameliorate MASH by restraining HSC proliferation and reducing excessive fibrosis [47]. The phenotypic changes of SASP concur with an increased presence of specific markers, such as altered levels of *P53* expression and up-regulation of cyclin-dependent kinase inhibitors, P21 and P16, which further characterize senescent liver cells [48].

Although the MASH mouse model used in the present studies does not progress to hepatocellular carcinoma (HCC), it is noteworthy that the activation of TGF-β and Hh signaling have been correlated with progression of MASH to HCC [49], and inhibition by Oxy210 may also have beneficial effects on HCC development, which we aim to study in the future using mouse models of MASH that do progress to cirrhosis and HCC [50]. Targeting TGF-β and, to a lesser degree, Hh signaling, has been considered a promising therapeutic approach for HCC since their inhibition can reduce tumor growth, increase sensitivity to other therapies, and improve clinical outcomes [51]. TGF-β is thought to play a dual role in HCC acting as a tumor suppressor in the early stages of the disease and then a tumor promoter during the later stages by supporting tumor cell survival, proliferation, and immune evasion [52,53]. Similarly, the role of Hh signaling appears to be complex [54,55,56] and potentially cell-type dependent [57].

In our WD-induced mouse model of MASH, hepatic manifestations begin after a few weeks, and the disease is fully developed in most animals after 16 weeks [14]. In mouse liver tissue harvested over the 16-week time course, 18 different pro-fibrotic, pro-inflammatory and senescence markers were induced by WD over time and most often effectively inhibited by Oxy210 treatment. One limitation of this study is exclusive reliance on mRNA expression. However, given the robustness of the results provided by mRNA data analysis, clearly demonstrating the elevated gene expression during the development of MASH and their significant lowering by Oxy210, these findings are still important and valuable for future research. In future studies, we plan to assess gene expression patterns associated with MASH development and the therapeutic effects of Oxy210 through metabolic biomarker and protein level analyses.

As liver tissue comprises a variety of cell types, such as hepatocytes, HSCs, Kupffer cells, etc., we cannot attribute our findings to any one specific cell type which is another limitation of this study. It is conceivable, for example, that increases in the pro-fibrotic gene expression induced by WD may originate mainly from HSCs, whereas increases in pro-inflammatory gene expression induced by WD may originate mainly from Kupffer cells, infiltrating macrophages, or other immune cells. Indisputably, activated HSCs and myofibroblasts, but not hepatocytes, are considered the primary source of liver collagen production in the context of MASH [8,36,58]. The specific WD-induced effects on hepatocytes are unclear but likely involve reduced cellular function and cell death, as suggested by increased plasma ALT levels, a measure of liver damage, and apoptosis in the liver observed in animals on WD both of which were ameliorated by Oxy210 [14,15]. Oxy210 treatment also resulted in significant lipid lowering effects in plasma and liver tissue that could potentially influence hepatic and vascular senescence [59]. Although total cholesterol levels were not reduced significantly in the liver but reduced in plasma by 48%, unesterified cholesterol levels were reduced both in the liver by 25% and in plasma by 37% [15,16]. This may suggest that the hepatic responses induced by Oxy210 are not limited to a minor reduction in hepatic cholesterol uptake from WD but likely include more significant effects on hepatic cholesterol synthesis, processing and trafficking as well.

Given the prominent role of TGF-β signaling in driving senescence in a disease agnostic manner, we propose that Oxy210 may inhibit senescence and associated fibrosis in models of other age-related diseases associated with aberrant TGF-β signaling. Thus far, we have studied the use of Oxy210, in vitro, in disease relevant human cells, for MASH and other fibrotic conditions, such as IPF and kidney fibrosis, as well as for atherosclerosis and cancer with encouraging results [15,16,20,60]. Expanding into meaningful in vivo proof of concept studies, however, can be challenging as in vivo models may not always recapitulate important aspects of human disease, including but not limited to cellular senescence [61].

## 5. Conclusions

We demonstrate significant increases in markers of senescence in parallel with fibrotic and inflammatory factors in the livers of APOE*3-Leiden.CETP mice during the development of MASH. Administration of Oxy210 significantly reduced the expression of these factors in a manner consistent with its potential as a therapeutic drug candidate for targeting MASH. Success of a therapeutic candidate for treating MASH is defined by the food and drug administration (FDA) and European Medicines Agency (EMA) as causing MASH resolution without worsening fibrosis, or improvement in fibrosis without worsening MASH. The triple inhibitory effects of Oxy210 on fibrosis, inflammation, and lipid deposition that occur during pathogenesis of MASH in mice, if translated in the future to humans, will have important clinical significance.

## Data Availability

The data presented in this study are available on request and within reason from the corresponding author (F.P.). The data are not publicly available due to privacy.

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
