# Peer review of "Oxy210 Inhibits Hepatic Expression of Senescence-Associated, Pro-Fibrotic, and Pro-Inflammatory Genes in Mice During Development of MASH and in Hepatocytes In Vitro"

_cells, 2025, doi:10.3390/cells14151191_

Round 1

Reviewer 1 Report

Comments and Suggestions for Authors

Manuscript : cells-3761193

Title: Oxysterol, Oxy210, inhibits hepatic expression of senescence- associated and pro-fibrotic genes in humanized hyperlipidemic  mice during development of MASH and in human hepatocytes  in vitro.

In this manuscript, the authors explored the impact of oxysterol Oxy210 on cellular senescence and fibrosis pathways, in tissues obtained from a mouse model of metabolic dysfunction-associated steatohepatitis and in a cell line of hepatocytes.

The paper is well written. Unfortunately, all the conclusions and claims rely in only one technique, real-time PCR. The senescence process requires a phenotypic characterization. This includes (but not restricted to) studying the same pathways, factors and proteins in situ, at the protein level. It is impossible to verify these phenomena based only on mRNA information.

Author Response

  1. Responses to comments by Reviewer 1:

Unfortunately, all the conclusions and claims rely on only one technique, real-time PCR. The senescence process requires a phenotypic characterization. This includes (but not restricted to) studying the same pathways, factors and proteins in situ, at the protein level. It is impossible to verify these phenomena based only on mRNA information.

We appreciate the concern by the Reviewer, and we agree that this is a limitation of our current manuscript. To address this concern, in the revised manuscript, we have included a statement in the Discussion Section highlighting this and other limitations of our study (please see, lines 352-366). Unfortunately, when the study was conducted by our collaborators, only RNA was made available for analysis, and at this time, we no longer have access to the tissue samples. However, given the robustness of the results provided by mRNA data analysis, clearly demonstrating the elevated expression of gene expression during the development of MASH and their significant lowering by Oxy210, we believe that reporting these findings is still important and valuable for future research by us and others in the field.

Reviewer 2 Report

Comments and Suggestions for Authors

The authors of this investigation still need to address the following issues before furthers steps.

  1. Sequences of primers were not submitted for the review process. Authors state that primers were submitted in supplemental Table 1, however, the table was not included anywhere.
  2. In the title, the sentences “humanized hyperlipidemic mice” and “human hepatocytes” are unnecessarily; actually, they sound as methodological procedures. The title should be attractive and summarize the main finding of the investigation rather than explaining methodological procedures.
  3. Although authors made a great effort to construct figure 7, most of the associated mechanisms/diseases shown in the figure were not investigated in the present original manuscript. Figure 7 is more appropriate for a review article than for this original research article. Authors should be aware that a graphical abstract of an original article, if that is what they intend to show with Figure 7, should only show mechanisms/phenomena directly associated with the conducted research. So, in the context of this research, figure 7 is too speculative since what it shows is not supported by the experimental evidence shown in the manuscript. Authors re encouraged to write a review article and include that figure.
  4. Gene and protein short names, after definition, shall be cited as their accepted symbols, and they shall be indicated according to the accepted nomenclatures. For example, the rule states that depending on the specie, symbols must be written using combination of lowercase and uppercase letters, italics, etc. Throughout the manuscript including text, tables, figures, figure legends, and supplementary information, gene and protein symbols shall be properly indicated. The author should know that both gene and protein short names are symbols, not abbreviations, and they are different among species. As reference, authors should review the HUGO Gene Nomenclature Committee guidelines and, GeneCards website, and read the following article: PMID: 22836666. For example, in result section 3.1, genes symbols were wrongly written as P21, P15, P16 and P53 although the studied specie was mouse.

Author Response

  1. Responses to comments by Reviewer 2:

1) Sequences of primers were not submitted for the review process. Authors state that primers were submitted in supplemental Table 1, however, the table was not included anywhere.

We apologize for this omission and the Table has now been submitted as Table 1 in the supplementary material.

2) In the title, the sentences “humanized hyperlipidemic mice” and “human hepatocytes” are unnecessarily; actually, they sound as methodological procedures. The title should be attractive and summarize the main findings of the investigation rather than explaining methodological procedures.

We agree with the Reviewer. Accordingly, we have changed the title of the revised manuscript to: Oxy210 Inhibits Hepatic Expression of Senescence-Associated, Pro-Fibrotic, and Pro-Inflammatory Genes in Mice During Development of MASH and in Hepatocytes In Vitro”.

3) Although authors made a great effort to construct figure 7, most of the associated mechanisms/diseases shown in the figure were not investigated in the present original manuscript. Figure 7 is more appropriate for a review article than for this original research article. Authors should be aware that a graphical abstract of an original article, if that is what they intend to show with Figure 7, should only show mechanisms/phenomena directly associated with the conducted research. So, in the context of this research, figure 7 is too speculative since what it shows is not supported by the experimental evidence shown in the manuscript. Authors re encouraged to write a review article and include that figure.

We agree with the reviewer. Accordingly, we have removed Figure 7 from the revised manuscript and added a Graphical Abstract instead.

4) Gene and protein short names, after definition, shall be cited as their accepted symbols, and they shall be indicated according to the accepted nomenclatures. For example, the rule states that depending on the species, symbols must be written using combination of lowercase and uppercase letters, italics, etc. Throughout the manuscript including text, tables, figures, figure legends, and supplementary information, gene and protein symbols shall be properly indicated. The author should know that both gene and protein short names are symbols, not abbreviations, and they are different among species. As reference, authors should review the HUGO Gene Nomenclature Committee guidelines and, GeneCards website, and read the following article: PMID: 22836666. For example, in results section 3.1, genes symbols were wrongly written as P21, P15, P16 and P53 although the studied species was mouse. We thank the reviewer for bringing this matter to our attention.

We thank the Reviewer for raising these important concerns and have made the appropriate corrections in the revised manuscript (see Figure 1).

Reviewer 3 Report

Comments and Suggestions for Authors

The manuscript titled ‘ Oxysterol, Oxy210, inhibits hepatic expression of senescence- 2

associated and pro-fibrotic genes in humanized hyperlipidemic 3mice during development of MASH and in human hepatocytes 4in vitro’ investigated the MASH inhibitory activity of the oxysterol derivative oxy210. their study found out  that it resulted in  reduced hepatic inflammation, lipid deposition, and fibrosis in mice. Even though the authors have shown good efforts to provide evidences confirming the therapeutic effects of Oxy210 in MASHthis manuscript may require minor revisions before its publication. Some specific comments are provided below.

1. The methodology part is too brief, it should include details on th rational behind the dose selection, references for methods and elaboration of procedures in some parts.

2. Authors may include introductory information about the oxy210 biochemical characteristics from its bioactivity perspective

3. Did they attempt to determine liver function of the mice by measuring serum level of liver function enzymes. This is a basic experiment when assessing and characterizing effects of bioactive compounds on liver pathophysiology

Author Response

  1. Responses to comments by Reviewer 3:

  1. The methodology part is too brief, it should include details on th rational behind the dose selection, references for methods and elaboration of procedures in some parts.

We appreciate this feedback by the Reviewer and have elaborated on details used for experimental design in the revised manuscript (see Section 2.1 and Section 2.3).

  1. Authors may include introductory information about the oxy210 biochemical characteristics from its bioactivity perspective.

We appreciate this feedback, and we have now included additional information regarding Oxy210 in the revised manuscript (Section 2.1 and Discussion lines 301-305).

  1. Did they attempt to determine liver function of the mice by measuring serum level of liver function enzymes. This is a basic experiment when assessing and characterizing effects of bioactive compounds on liver pathophysiology.

We respectfully point out that a tight correlation between plasma alanine transaminase (ALT), a measure of  liver health and function, and the degree of liver fibrosis and MASH in the mouse model employed in the present manuscript was previously established and reported by our coauthors, Drs Hui and Lusis (reference 14, PMID: 29907965).   We also reported on the beneficial effects of Oxy210 on lowering plasma ALT levels by comparing plasma ALT levels in the Western Diet (WD) cohort that develops MASH and fibrosis with the WD + Oxy210 group that showed significant lowering of ALT with Oxy210 treatment (reference 15, PMID: 34505423).
